# The Micromorphology and Its Taxonomic Value of the Genus *Sanicula* L. in China (Apiaceae)

**DOI:** 10.3390/plants13121635

**Published:** 2024-06-13

**Authors:** Boni Song, Feng Yong, Changkun Liu, Yunyi Wang, Lei Yang, Lian Chen, Yuan Wang, Songdong Zhou, Xingjin He

**Affiliations:** 1Key Laboratory of Bio-Resources and Eco-Environment of Ministry of Education, College of Life Sciences, Sichuan University, Chengdu 610065, China; songboni@stu.scu.edu.cn (B.S.); Wangyy062021@163.com (Y.W.); yx1004821224@163.com (L.Y.); lianchen102499@163.com (L.C.); wang_yuan2000@163.com (Y.W.); zsd@scu.edu.cn (S.Z.); 2School of Life Science and Engineering, Lanzhou University of Technology, Lanzhou 730050, China; lut8866@163.com; 3College of Resources Environment and Chemistry, Chuxiong Normal University, Chuxiong 675000, China; liuchangkun@cxtc.edu.cn

**Keywords:** *Sanicula* L., leaf epidermis, fruit, pollen, ultrastructural taxonomy

## Abstract

The genus *Sanicula* L. possesses many medically important plants, belonging to the family Apiaceae. It is one of the most taxonomically difficult taxa, largely due to the great variability in habit, foliage, flowers and fruits. Previous studies have mainly focused on the molecular studies of this genus, and the morphological research for this genus was limited, especially in the micromorphological research. In the current study, we newly obtained leaf materials from twenty-two *Sanicula* members, fruit and pollen materials from twenty *Sanicula* members and performed comprehensively micromorphological analyses for this complicated genus. The results of the leaf epidermis showed that the upper and lower epidermis were smooth and glabrous, and the cell shape was polygonal or irregular. The patterns of anticlinal wall were shallowly undulating, deeply undulating, subflat or flat. The cuticular membrane ornamentations were diverse, and some species had epidermal appendage. All *Sanicula* species observed the stomata in the lower epidermis, and only five species (*S. rugulosa*, *S. elongata*, *S. hacquetioides*, *S. tienmuensis* and *S. elata*) observed stomata in the upper epidermis, which can easily identify them from other *Sanicula* members. In addition, we found that the fruits scarcely compressed, and some fruits had their distinctive shape, such as the fruit shape of *S. tienmuensis* was subglobose, *S. subgiraldii* was broadly ovate and *S. pengshuiensis* was ellipsoid. All *Sanicula* taxa fruits surfaces were covered with prickles, bristles, protuberance, or tubercles, prickles were either long or short, uncinate or straight, rarely scale-like, ribs inconspicuous or slightly prominent, but the prickles/bristles/tubercles were different in shape, sparseness and arrangement. The vittae were distinct in *S. rubriflora*, *S. chinensis*, *S. caerulescens*, *S. pengshuiensis*, *S. pauciflora*, *S. lamelligera*, *S. oviformis*, *S. flavovirens* and *S. elata*, and the remaining taxa were obscure. These findings indicated that the fruits can clearly distinguish these *Sanicula* members. Furthermore, the micromorphological characteristics of pollen showed that the equatorial view included four shapes: ellipsoid, subrectangular, equatorially constricted and super-rectangular-equatorially constricted; and the polar view possessed four shapes: triangular, triangular–circular, suborbicular and trilobate circular. The germ furrow and the outer wall ornamentation of all *Sanicula* taxa were quite similar, indicating that the genus was a natural unit. In summary, our study promoted the improvement of a taxonomic system for the genus and also provided additional evidence for future taxonomic study of the genus *Sanicula*.

## 1. Introduction

*Sanicula* L. (Apiaceae-Saniculoideae) contains about 45 species worldwide that are widely distributed from East Asia to North America [1,2], with China and North America as two distribution centers [3]. Of those, nineteen species and five varieties are distributed in China, with eleven species and five varieties of them endemic [4,5,6,7]. The genus was considered as a taxonomically difficult group within Apiaceae and exhibited diversity in habits, leaves, inflorescences, fruits and rhizomes [2,3]. The plants of the genus mainly grow in grassy places on stream banks, rocky walls, alpine scrub, shady wet places, mixed forests on mountain slopes, bamboo forests, wet shady valleys and other habits [5,6] (Figure 1). The members of the genus are biennial or perennial herbs, and the stems are erect, ascending or rarely decumbent. Interestingly, the stems of Chinese *Sanicula* species are glabrous [5,6] (Figure 2). Their leaves are varied, mainly including blade orbicular, round–cordate or cordate–pentagonal, membranous, papery, or subleathery, trifoliolate, palmately 3-parted or 3–5-parted and often lobed, margin serrate or doubly setose serrate. For example, the basal leaves of *S. caerulescens* Franch. are trifoliolate; *S. rubriflora* F. Schmidt, *S. rugulosa* Diels and *S. hacquetioides* Franchet. are palmately 3-parted; *S. giraldii* H. Wolff and *S. elongata* K. T. Fu are palmately 3–5-parted [5,6,8,9] (Figure 3). The genus has its distinctive inflorescences in that their umbels are simple or in irregularly elongated compound umbels, peduncles racemous, cymous or corymbose-branched. For example, *S. caerulescens* Franch. has inflorescence subracemose, sometimes several umbels in fascicles; *S. orthacantha* S. Moore has inflorescence 2–3-branched, umbels 3–8 and *S. hacquetioides* Franchet. has inflorescence terminal [5,6,8,9] (Figure 4). The fruit of the genus also has its unique characteristic features (conspicuously prominent and persistent calyx, two persistent styles and fruit covered with scales, bristles or hooked prickles). All these distinctive characteristics can easily distinguish the genus *Sanicula* from other genera of the Apiaceae [10,11].

Since 1975, Behnke proposed the concept of “ultrastructural taxonomy” [12], opening up a new research field for the search for extensive evidence in taxonomy. Micromorphology has become an important tool for studying the evolutionary relationships and the taxonomy of plant systems, especially for those taxonomically notorious groups, such as *Impatiens* L. [13], *Camellia* L. [14], *Yulania* Spach [15], *Pugionium* Gaertn. [16], as well as the genera of Apiaceae [17,18,19,20,21,22]. In recent years, some scholars have also performed micromorphological studies on the genus *Sanicula* [23,24,25,26,27,28,29,30,31,32]. For example, Ma et al. observed the leaf epidermis of six *Sanicula* species under a light microscope and found that these species had the same type of epidermal structure [23]. Chen et al. observed the fruit surface micromorphology of fifteen *Sanicula* plants (only including eight Chines taxa) and detected that the fruit size, degree of curvature of prickles, and the wax pattern of the fruit stalks of plants in this genus were rich and diverse [29]. Yang et al. observed the pollen micromorphology of fifteen *Sanicula* species and concluded that the pollen of the genus *Sanicula* had moderately evolved and relatively evolved morphological features [32]. Therefore, it can be seen that micromorphological characteristics (for example: leaf epidermis, fruit and pollen) of species are useful for distinguishing morphologically similar or taxonomically difficult species, as well as providing more additional evidence for these species. Although previous micromorphological studies have greatly filled the gaps and provided valuable references for the genus *Sanicula*, the relationships of species with taxonomic difficulties within the genus have not been satisfactorily resolved because these studies involved fewer species (especially the lack of Chinese endemic species) and did not conduct a comprehensive study. Therefore, it is urgent to further explore the micromorphological features of this genus based on expanding sampling.

In this study, we obtained leaf materials from twenty-two species, fruit and pollen materials from twenty species in which these taxa have similar morphology and species identification is difficult, and we performed comprehensively micromorphological analyses of this taxonomically difficult genus. Our major aims were to: (1) reveal the micromorphology of leaf epidermis, fruit and pollen of the genus; (2) evaluate the potential of micromorphology for resolving the taxonomy; (3) provide more additional evidence for exploring the interspecific relationship of the genus.

## 2. Material and Method

### 2.1. The Micromorphological Study on the Leaf Epidermis of the Genus Sanicula *L.*

The leaves materials used in this study were collected in the wild. To avoid errors in experimental materials, we selected clean and fresh basal leaves from each species and then immediately dried the leaves with silica gel and brought the dried leaves back to the laboratory for experiments. Details of the material sources and voucher specimens were shown in Appendix A, and the voucher specimens were deposited in the Sichuan University Herbarium (SZ). We used a light microscope and scanning electron microscope to observe the leaf epidermal micromorphological features, and the terminology related to leaf epidermal features referred to the studies of Baranova [33], Wilkinson [34], Zhang et al. [19] and Xu et al. [35].

(1)Light microscope observation: Silica gel-dried basal leaves were placed in 30% sodium hypochlorite solution (NaClO_3_) in the dark treatment for 1–3 h. When the leaves faded to white and were nearly transparent, we removed the leaves and immersed them in distilled water for 5–10 min. Then, the leaves were removed and placed on a glass plate, and we gently tore the thin and transparent upper and lower epidermis with tweezers and placed them on a slide with a drop of distilled water to unfold, covered with a coverslip, gently pressed to squeeze out the excess water and air under the coverslip, and the excess water was carefully sucked off with absorbent paper to make a temporary slide. Finally, under 10 × 40 magnification, we respectively selected six fields of view with high clarity of the upper and lower epidermis, used a light microscope (Olympus-BX51) to observe the cells and stomata of the leaf epidermis, and photographed and preserved them. We randomly selected thirty stomata in the lower epidermis of each taxon, and due to the small number of stomata in the upper epidermis of some species, all stomata were selected. Then, we observed the cell shape, the type of stomatal apparatus of both the upper and lower epidermis, counted the number of cells, measured the size of the stomatal apparatus, and calculated the stomatal density and stomatal index. Stomatal density: number of stomata per unit area/unit area. Stomatal index: number of stomata per unit area/(number of stomata per unit area + number of epidermal cells in the same unit area) × 100%. The above measurement processes were carried out in MATO v4.2 software [36].(2)Scanning electron microscope observation: The silica gel-dried basal leaves were washed with distilled water and then treated with different concentrations of ethanol for gradient dehydration (15%, 30%, 50%, 70%, 85%, 95% Ⅰ, 95% Ⅱ, 100% Ⅰ, 100% Ⅱ), with each concentration being processed for 1 min. Then, we took out the leaves and dried them naturally. Subsequently, the upper and lower epidermis of each species were cut out and were pasted on double-sided conductive adhesive and sprayed with gold coating, respectively. Finally, we observed and photographed the micromorphological features of the upper and lower epidermis using a JSM-7500F scanning electron microscope.

### 2.2. The Micromorphological and Anatomical Study on the Fruits of the Genus Sanicula *L.*

The fruit materials used in this study were collected in the wild. First, we collected the mature and intact fruits and then immersed them in formaldehyde–acetic acid–alcohol (FAA) fixation fluid (38% formaldehyde: glacial acetic acid: 70% ethanol: glycerol = 1:1:18:1). Fruit materials were stored in the Laboratory of Plant Taxonomy and Systematic Evolution, School of Life Sciences, Sichuan University, and the information of fruit material sources and voucher specimens were shown in Appendix A. The voucher specimens were deposited in the Sichuan University Herbarium (SZ). The terminology for the description of micromorphological and anatomical characteristics of fruits referred to the studies of Kljuykov [37] and Ostroumova et al. [38].

(1)Observation of external micromorphological characteristics of fruits. We removed the fruits from the FAA fixation fluid and placed them on absorbent paper to allow them to dry naturally. Then, ten morphologically intact fruits of each species were randomly selected for observation under a stereomicroscope (SMZ25, Nikon Corp., Tokyo, Japan), and then the dorsal and commissure views of fruits of each species were observed, photographed and preserved. Finally, the length and width values of ten fruits of each species were measured using MATO v4.2 software [36], and we then calculated the mean values.(2)Observation of anatomical characteristics of fruits.a:Freehand slicing method. We observed and randomly selected ten morphologically intact fruits of each species using a stereomicroscope (SMZ25, Nikon Corp., Tokyo, Japan). Then, we used double-layer blades to cut the fruits in the vertical horizontal plane at the middle position of the fruits. Finally, we observed and photographed the transverse section of the fruits with a stereomicroscope (SMZ25, Nikon Corp., Tokyo, Japan).b:Paraffin sectioning method. We observed and randomly selected one or two morphologically intact fruits of each species and placed them in FAA fixation fluid for 2–24 h. After the fixation process was completed, we removed them and washed them with distilled water 2–3 times. Then, the fruits were subjected to gradient dehydration treatment with ethanol (30%, 50%, 70%, 80%, 90%, 100%) for 1 h at each concentration. After the gradient dehydration process was completed, we used xylene for deethanolization for 30 min. Finally, we permeabilized, embedded, sectioned, glued, dewaxed the treated fruit materials and stained them using toluidine blue solution (Toluidine Blue solution), and meanwhile, the slices were sealed with neutral gum to create permanent mounts. The anatomical features of the fruits (such as the vittae, endosperm, exocarp and mesocarp) were observed and photographed using a stereomicroscope (SMZ25, Nikon Corp., Tokyo, Japan).
(3)Observation of micromorphological characteristics of fruits: We randomly selected one or two morphologically intact fruits of each species from the FAA fixation fluid. The remaining steps referred to the scanning electron microscopy method for leaf epidermis (Section 2.1 (2)).

### 2.3. The Morphological Study on the Pollen of the Genus Sanicula *L.*

We collected the mature, dry and full pollen materials used in this study in the wild, which were naturally dried and used for subsequent experiments. Details of pollen material sources and voucher specimens were shown in Appendix A, and the voucher specimens were deposited in the Sichuan University Herbarium (SZ). The pollen morphological features were observed using a light microscope and scanning electron microscope, and the terminology related to pollen morphological features referred to the studies of Wang et al. [39], She and Shu [40,41] and Yang et al. [32].

(1)Light microscope observation: The G. Erdtman acetic anhydride decomposition method [42] was used to treat pollen, and the specific steps were as follows:a:Take an appropriate amount of mature, dry and full pollen of each species in a 5 mL cryostat tube and submerge the pollen with 1–2 mL of glacial acetic acid for 24 h;b:Use a glass rod to thoroughly crush the pollen in the cryostat tube, filter impurities with a 100-mesh copper mesh, wash with glacial acetic acid and collect 5 mL of filtrate into a new cryostat tube;c:Put the new cryostat tube in a centrifuge at 5000 r/min for 10 min and then repeat again;d:Add a mixture of acetic anhydride and concentrate sulfuric acid (volume ratio = 9:1, currently used and prepared) into the cryostat tube containing discarded supernatant and add dropwise to 5 mL;e:Place the frozen tube in a water bath at 80 °C for 4 min, then remove and centrifuge at 5000 r/min for 10 min;f:Discard the supernatant from the frozen tube, add 4 mL of distilled water and centrifuge at 5000 r/min for 10 min;g:Discard the supernatant from the freezing tube, then add 2–3 drops of glycerol to the freezing tube and mix well;h:Pipette 1–2 drops of pollen suspension onto a glass slide and create a temporary slide.


Firstly, we searched for the pollen with intact morphology under low magnification. Then, we randomly selected twenty pollen of each species to observe and photograph their shape and the equatorial and polar views under a light microscope (Olympus-BX51) at 10 × 100 magnification. Finally, MATO v4.2 software [36] was used to measure the pollen size, including the polar axis length (P) and the equatorial axis length (E) of pollen. In addition, we also calculated the ratio of polar axis length and equatorial axis length (P/E).

(2)Scanning electron microscope observation: Firstly, we observed and randomly selected ten dry, mature and full anthers of each species under a dissecting microscope. Then, the selected anthers were pasted on the double-sided conductive adhesive, and the pollen capsule was gently poked with a clean dissecting needle to release the pollen grains, so that they were evenly spread in all areas of the conductive adhesive, and the shell of the pollen was clamped away with tweezers. Finally, after gold spray coating, we used a JSM-7500F scanning electron microscope to observe and photograph the overall view of pollen, the polar view, the equatorial view, germ furrow and the exine ornamentation.

## 3. Results

### 3.1. The Morphological Characteristics of the Leaf Epidermis

#### 3.1.1. Leaf Epidermal Characteristics by Light Microscope

We used a light microscope to observe the micromorphological characteristics of the leaf epidermis of twenty *Sanicula* plants, of which the eleven taxa were reported for the first time (Figure 5).

Our study observed that the upper and lower epidermis were smooth and glabrous. For the upper epidermis, the cell shape of these twenty *Sanicula* taxa can be divided into two types: polygonal and irregular. Among them, the cell shape in the upper epidermis of *S. chinensis*, *S. elongata*, *S. brevispina*, *S. subgiraldii*, *S. pengshuiensis*, *S. pauciflora*, *S. oviformis*, *S. tienmuensis*, *S. flavovirens* and *S. elata* were polygonal, and the cell shape of the remaining taxa were irregular. The pattern of anticlinal wall had three types: shallowly undulating, deeply undulating and subflat or flat. Of which, the pattern of the anticlinal wall of *S. brevispina* and *S. orthacantha* was shallowly undulating; *S. elongata*, *S. subgiraldii*, *S. pengshuiensis*, *S. pauciflora*, *S. oviformis*, *S. tienmuensis*, *S. flavovirens*, *S. elata* and *S. nanchuanensis* were subflat or flat, and the remaining taxa were deeply undulating. The stomata were only observed in *S. rugulosa*, *S. elongata*, *S. hacquetioides*, *S. tienmuensis* and *S. elata*, and the stomatal shape was ellipsoid in *S. rugulosa*, *S. hacquetioides*, *S. tienmuensis*, *S. elata* and suborbicular in *S. elongata*. As for stomatal size (length × width), the largest was 32.43 μm × 19.73 μm in *S. rugulosa*, and the smallest was 22.81 μm × 22.49 μm in *S. elongata*. The largest stomatal ratio was 2.31 for *S. elata*, and the smallest was 1.01 for *S. elongata*. The largest stomatal density and stomatal index were observed for *S. rugulosa* (82.54/mm^2^, 15.52%), and the smallest was for *S. elata* (16.18/mm^2^, 1.30%) (Figure 5(A1–T1), Appendix A).

For the lower epidermis, the cell shape of these twenty *Sanicula* taxa were irregular. The pattern of the anticlinal wall had two types: shallowly undulating and deeply undulating. For instance, *S. chinensis*, *S. elongata*, *S. subgiraldii*, *S. pauciflora*, *S. oviformis*, *S. tienmuensis*, *S. elata* and *S. nanchuanensis* were shallowly undulating, and the remaining taxa were deeply undulating. We observed that the stomata existed in these twenty *Sanicula* taxa. The stomatal shapes were ellipsoid, long-ellipsoid, ovoid and suborbicular. Among them, *S. tienmuensis* had the suborbicular stomata; *S. serrata*, *S. pauciflora*, *S. hacquetioides*, *S. orthacantha* and *S. flavovirens* had the ellipsoid stomata; *S. chinensis*, *S. subgiraldii*, *S. oviformis*, *S. elata* and *S. nanchuanensis* had the ovoid stomata; and the remaining taxa had the ellipsoid stomata. The stomatal size (length × width) was 27.20 × 18.49 μm–20.03 × 17.90 μm, and the largest stomata was observed in *S. serrata*, and the smallest stomata was for *S. nanchuanensis*. The stomatal ratio ranged from 1.12 (*S. nanchuanensis*) to 1.56 (*S. orthacantha*). The stomatal density of *S. oviformis* was the largest (681.16), and that of *S. pengshuiensis* was the smallest (81.09). The stomatal index ranged from 8.46 (*S. pengshuiensis*) to 29.41 (*S. rugulosa*) (Figure 5(A2–T2), Appendix A).

#### 3.1.2. Leaf Epidermal Characteristics by Scanning Electron Microscope

We used a scanning electron microscope to observe the micromorphological characteristics of the leaf epidermis of twenty-two *Sanicula* plants, of which the leaf epidermis of sixteen members were reported for the first time (Figure 6 and Figure 7).

In the upper epidermis of these twenty-two *Sanicula* plants, we observed that the cuticular membrane ornamentation can be divided into four types: sparsely striated and clustered into protuberances, densely striated and clustered into protuberances, sparsely striated and striated protuberance, densely striated and striated protuberance. The cell contour was visible in eight taxa (*S. giraldii*, *S. caerulescens*, *S. serrata*, *S. brevispina*, *S. subgiraldii*, *S. oviformis*, *S. nanchuanensis*, *S. petagnioides*), and the cell contour was invisible in the other fourteen taxa. Of these, five species (*S. caerulescens*, *S. serrata*, *S. elongata*, *S. hacquetioides* and *S. pauciflora*) had no epidermal appendage, and the remaining seventeen species had epidermal appendages. The stomata were only observed in *S. rugulosa*, *S. elongata*, *S. hacquetioides*, *S. tienmuensis* and *S. elata* (Figure 6, Appendix A).

In addition, in the lower epidermis of these twenty-two *Sanicula* plants, the cuticular membrane ornamentation features were smooth, nearly smooth, sparsely striated and striated protuberance, sparsely striated and clustered into protuberances. The cell contour was invisible in all these *Sanicula* taxa. The epidermal appendage was absent in five species (*S. caerulescens*, *S. serrata*, *S. hacquetioides*, *S. orthacantha*, *S. brevispina*), and the remaining seventeen species possessed epidermal appendage. All members had stoma, and the stoma included two types: protuberant and sunken. Among them, the stoma of ten species (*S. rugulosa*, *S. giraldii*, *S. caerulescens*, *S. serrata*, *S. hacquetioides*, *S. orthacantha*, *S. oviformis*, *S. tienmuensis*, *S. flavovirens*, *S. nanchuanensis*) was sunken, and the stoma was protuberant in the remaining twelve species (Figure 7, Appendix A).

### 3.2. The Micromorphological and Anatomical Characteristics of the Fruits

We observed twenty *Sanicula* fruits, and the fruits micromorphological characters of thirteen taxa were reported for the first time (Figure 8, Figure 9 and Figure 10). We found that the fruit colors were brownish-yellow, brownish-brown or purplish-red. The fruits scarcely compressed, and the fruit shapes were diverse, such as ovoid or ovoid–globose, globose or ellipsoid, broadly ovate and subglobose, of which some *Sanicula* members fruits had their distinctive shape, such as the fruit shape of *S. tienmuensis* was subglobose; *S. subgiraldii* was broadly ovate; *S. pengshuiensis*, *S*. *rugulosa* and *S. nanchuanensis* were ellipsoid, *S. elongata*, *S. oviformis* and *S. brevispina* were ovoid; *S*. *rubriflora*, *S*. *chinensis*, *S*. *serrata*, *S*. *hacquetioides*, *S*. *flavovirens* and *S*. *elata* were ovoid–globose. The fruit size varied from 1.89 × 0.90 mm (*S. astrantiifolia*) to 6.97 × 2.68 mm (*S. rubriflora*). The fruit surface was covered with prickles or protuberances, or tubercles, prickles were either long or short, straight or hooked, rarely scale-like. Among them, *S. astrantiifolia*, *S. chinensis*, *S. giraldii* and *S. rugulosa* had the extremely well-developed uncinate prickles; *S. flavovirens*, *S. pauciflora* and *S. orthacantha* had the straight bristles; *S. hacquetioides*, *S. oviformis* and *S. tienmuensis* were covered with scales and tubercles, but never spinulose; *S. caerulescens*, *S. elongata*, *S. pengshuiensis* and *S. lamelligera* were covered with short and straight spinous-bristles, usually fused at the base forming a thin tier; *S. nanchuanensis* were proximal end with degenerated-to-disappeared prickles, nearly smooth, distal end with prickles and formed a thin layer; *S. rubriflora*, *S. serrata* and *S. subgiraldii* had their proximal part covered with scales, distal part covered with slightly uncinate bristles. The calyx teeth were prominent in all *Sanicula* taxa, and the shape was lanceolate, ovate and linear. Of them, *S. giraldii*, *S. serrata* and *S. tienmuensis* had ovate calyx teeth; *S. chinensis*, *S. pengshuiensis* and *S. lamelligera* had linear calyx teeth; and the remaining taxa had the varied shape of these three shapes (Figure 8, Table 1).

In addition, we also detected the micromorphological characteristics of fruits surface, and we found that the cell shapes of fruit furrows were diverse, including circular shape, polygon shape and invisible. For example, *S. rubriflora*, *S. giraldii*, *S. flavovirens*, *S. tienmuensis* and *S. nanchuanensis* had the polygon cell shape; *S. elongata*, *S. brevispina* and *S. oviformis* had the circular cell shape. The surfaces of fruit furrows had various types, such as high convex, immersed, concave–convex and smooth (Figure 9).

Furthermore, exocarp and mesocarp were protruding as prickles or protuberances. The mericarp were suborbicular in the transverse section, sparsely ellipsoid–orbicular or subovoid. The endosperm on the commissural side was slightly concave or flat, which ten species (*S. rubriflora*, *S. caerulescens*, *S. serrata*, *S. elongata*, *S. brevispina*, *S. lamelligera*, *S. oviformis*, *S. tienmuensis*, *S. rugulosa* and *S. subgiraldii*) had the slightly concave endosperm on the commissural side, and the other ten species had the flat endosperm on the commissural side (Table 1). The exocarp consisted of 1 layer of slightly compressed or irregularly rounded thin-walled cells, the outer wall often with papillate projections. The prickles often contained lignified longitudinal elongated thick-walled tissue. The vittae were distinct in *S. rubriflora*, *S. caerulescens*, *S. pengshuiensis*, *S. pauciflora*, *S. lamelligera*, *S. oviformis* and *S. flavovirens*, and the vittae of the remaining taxa were obscure (Figure 8 and Figure 10).

### 3.3. The Micromorphological Characteristics of Pollen

We observed micromorphological characteristics of pollen from twenty *Sanicula* taxa, and the pollen micromorphological characteristics of seven taxa were reported for the first time (Figure 11 and Figure 12). The pollen size (polar axis × equatorial axis) ranged from 30.03 × 16.95 μm to 57.18 × 23.59 μm, with the smallest pollen in *S. chinensis* and the largest in *S. flavovirens*. The ratio (P/E) of the polar axis (P) to the equatorial axis (E) ranged from 1.58 to 2.65, with the minimum value occurring in *S. oviformis* and the maximum value occurring in *S. pauciflora*. The equatorial view of the pollen had four shapes: ellipsoid, subrectangular, equatorially constricted, super-rectangular-equatorially constricted. Of these, the equatorial views of *S. giraldii*, *S. elata* and *S. lamelligera* were subrectangular; *S. caerulescens*, *S. nanchuanensis* and *S. pauciflora* were equatorially constricted; *S. flavovirens* was superrectangular-equatorially constricted, and the remaining taxa had the elliptical equatorial view. The polar view of the pollen had four shapes: triangular–circular, triangular, suborbicular and trilobate circular. Among them, *S. giraldii*, *S. caerulescens* and *S. orthacantha* had suborbicular polar view; *S. rugulosa* and *S. astrantiifolia* had triangular polar view; *S. subgiraldii*, *S. pengshuiensis*, *S. lamelligera*, *S. elata*, *S. flavovirens* and *S. nanchuanensis* had trilobate circular polar view; and the remaining species had triangular–circular polar view. Except for the exine ornamentation of the equatorial view in *S. rubriflora* was brain-striped, and the remaining species had striped exine ornamentation in both their equatorial view and polar view. All *Sanicula* members had a side germinal aperture, and the germinal furrows were all long (Figure 11 and Figure 12, Appendix A).

## 4. Discussion

### 4.1. The Micromorphological Characteristics and Its Taxonomic Value of Leaf Epidermis in the Genus Sanicula *L.*

The micromorphology of plant leaf epidermis can reflect species interspecific relationships and affinities of species to some extent, which can provide a valuable reference for taxonomic issues, such as species identification and delimitation [43,44]. Therefore, leaf epidermal micromorphological features have become one of the most important evidence for plant taxonomic studies [45,46]. In recent years, the micromorphological characteristics of plant leaf epidermis have been widely used in the classification and systematic evolution of Apiaceae plants [5,6,7,8,9,10,47,48,49,50,51]. For example, Zhou et al. used light microscopy and scanning electron microscopy to study the leaf epidermis morphology of 21 *Peucedanum* L. plants, revealing that the leaf epidermis morphology of 21 *Peucedanum* L. plants was stable at the species level, and it can be used for species identification; however, there was almost no consistency at the group level, suggesting that the genus *Peucedanum* L. may not be a natural group [47]. Xu et al. observed the leaf epidermis micromorphology of seven *Phytophermopsis* H. Wollf plants using scanning electron microscopy and found that the contours of leaf epidermal cells, cell shape, ornamentation of cuticle and the stoma were different among these seven species. For example, the leaf epidermal cells of *P. muliensis* were regular polygonal, and the ornamentation of cuticle had irregular stripes. The stoma of *P. delavayi*, *P. shaniana*, *P. rubrinervis* and *P. obtusiuscula* were only distributed in the lower leaf epidermalon. The shape of the stoma was elliptical or ovate. Thus, these seven *Physospermopsis* species can be distinguished based on their micromorphological characteristics of leaf epidermis [35]. Ren et al. observed the leaf epidermal micromorphology of eleven *Pleurospermum* Hoffm. plants and found that the distribution of stoma was a valuable taxonomic evidence for these *Pleurospermum* taxa, and they also speculated that *P. franchetianum* and *P. davidii* had a close relationship [50].

Our study found that the micromorphological characteristics of leaf epidermis have high consistency within the genus *Sanicula*. For instance, the cell shape in the lower epidermis was irregular in all taxa, and the corresponding pattern of the anticlinal wall was shallowly or deeply undulating. The stomata was presented in the lower epidermis of all species. These findings suggested that the genus *Sanicula* was a natural taxon, which was consistent with the previous studies [13,14].

Based on the cell shape in the upper epidermis, we can easily distinguish eleven species (*S. chinensis*, *S. elongata*, *S. brevispina*, *S. subgiraldii*, *S. pengshuiensis*, *S. pauciflora*, *S. oviformis*, *S. tienmuensis*, *S. flavovirens*, *S. nanchuanensis* and *S. elata*) from other species because their cell shape was polygonal, while the cell shape of the remaining taxa was irregular. We can also identify five species (*S. rugulosa*, *S. elongata*, *S. hacquetioides*, *S. tienmuensis* and *S. elata*) from other species by the stomata because the stomata were only observed in these five members, and the other members had no stomata. Li et al. had treated *S. pengshuiensis* as a synonymy of *S. lamelligera* [52]. However, we did not support this view based on the study of leaf epidermal micromorphology. The cell shape of the upper epidermal in *S. pengshuiensis* was polygonal, while *S. lamelligera* was irregular, whereby they can be easily distinguished. In addition, the stomata in the lower epidermis of *S. pengshuiensis* was ovoid, while that in *S. lamelligera* was elliptical, so we considered them as two independent species. The previous molecular research also justified our micromorphological study [3]. Song et al. performed phylogenetic analyses based on plastome data and found that *S. tienmuensis* and *S. astrantiifolia* had close relationship with weak supports and low resolutions, which made them difficult to distinguish [3]. However, our study found that these two species had their own distinctive micromorphological features. For example, the upper epidermis of *S. astrantiifolia* had no stomata, the cell shape was irregular, and the cuticular membrane ornamentation was densely striated and striated protuberance, whereas the upper epidermis of *S. tienmuensis* had the ellipsoid stomata, the cell shape was irregular polygonal, and the cuticular membrane ornamentation was sparsely striated and clustered into protuberances. In addition, the stomata shape in the lower epidermis was also different, with ellipsoid stomata existing in *S. astrantiifolia*, whereas *S. tienmuensis* had a suborbicular stomata. From the above, it was noticed that *S. tienmuensis* and *S. astrantiifolia* had differences in the micromorphological characteristics of leaf epidermis, which can clearly distinguish them. We also found that three varieties (*S. subgiraldii*, *S. brevispina*, *S. pauciflora*) had distant affinities with their type varieties. In detail, *S. giraldii* and *S. orthacantha* had an irregular cell shape, while *S. subgiraldii* and *S. brevispina* had a polygonal cell shape in the upper epidermis. *S. tienmuensis* had an ellipsoid stomata, whereas *S. pauciflora* had no stomata in the upper epidermis. These findings implied that these three varieties should be regarded as three independent species. Furthermore, the stomatal size, stomatal ratio, stomatal index and stomatal density of epidermis had differences among these *Sanicula* taxa involved in this study, which can also serve as additional evidence to distinguish them. Therefore, it can be seen that the micromorphological characteristics of leaf epidermis can serve as a supplement to molecular data, providing new evidence for interspecific classification research of the genus *Sanicula*.

### 4.2. The Micromorphological Characteristics and Its Taxonomic Value of Fruit in the Genus Sanicula *L.*

The micromorphological characteristics of fruits play a vital role in the classification and systematic evolution of angiosperms at different levels of family, genus and species [53]. Especially, the characteristics of fruits in the family Apiaceae are very distinctive and stable, which have been widely used in the taxonomic research of this family [54,55,56,57,58,59]. For example, Liu et al. observed the fruit micromorphology of eighteen representative genera in Apiales and found that the anatomical characteristics of fruits were closely related to the systematic position of these genera, indicating that the anatomical characteristics of fruits were of high value in solving the phylogenetic and taxonomic problems of Apiales [58]. Kljuykov et al. enumerated fifteen features of the anatomical characteristics in the family Apiaceae and concluded that the anatomical characteristics of the fruit were usually constant in the genus, which can be used to describe the genus or closely related genera or groups [59]. Zhang et al. used a freehand sectioning method to observe and compare the external morphology and anatomical features of the fruits of 23 species in 17 genera of the family Apiaceae. They concluded that the characteristics of the fruit surface with hook spines or bristles and the fruit ribs showed significant differences among genera but showed consistency within genera, which can be used as the reference for the classification of the Apiaceae between genera. They also found that the shape of the fruit transverse section, the concavity and convexity of the endosperm surface, and the morphological characteristics of the calyx teeth were important in research studies on intragenus and interspecies classification. Based on the morphological characters of the fruit, a taxonomic search table for 23 plant species belonging to 17 genera was compiled [30].

We also observed the micromorphological and anatomical characteristics of the genus *Sanicula.* The most distinctive characteristic features of the genus *Sanicula* were the fruits covered with scales, bristles, or hooked prickles, a rather prominent and persistent calyx, and two persistent styles that can easily distinguish it from other genera of Apiaceae, which were in accordance with the previous studies [5,6,10]. In this study, twenty *Sanicula* species showed some differentiation in fruit micromorphology, and their characters reflected the similarities and differences among different species. Some *Sanicula* members fruits had their distinctive shape, prickles shape and calyx teeth. In addition, the cell shapes of the fruits surface were diverse, and the surface of fruit furrows had various types, which can also serve as important identified features among these *Sanicula* members. Therefore, the micromorphological characteristics of the *Sanicula* fruit had a certain taxonomic value and can be used as the basis for taxonomic identification. For example, the fruit size of *S. rubriflora* was significantly larger than other *Sanicula* species and can be easily distinguished from other members. The fruit shape of *S. tienmuensis* was subglobose, which was clearly different from other *Sanicula* species fruit. *S. hacquetioides* was easily distinguishable from other species because of its fruit covered with scales and tubercles, but never spinulose. *S. orthacantha* can be distinguished by its developed straight bristles, while *S. astrantiifolia*, *S. chinensis*, *S. giraldii* and *S. rugulosa* had the extremely well-developed uncinate prickles. *S. nanchuanensis* were proximal end with degenerated-to-disappeared prickles, nearly smooth, distal end with prickles and formed a thin layer. In addition, we also found that three varieties had distant affinities with their type varieties from fruits. For example, *S. giraldii* had fruit densely covered with developed yellow or purplish-red uncinate bristles, long and hard, calyx teeth ovate and small, tip mucronate, while *S. subgiraldii* had the unique fruits rarely covered with purplish-red short bristles, proximal end with tubercles, obscure, distal end with uncinate bristles or straight, calyx teeth broadly ovate and large. *S. tienmuensis* had fruits that were subglobose, covered with short and obtuse prickles, slight formed scales and tubercles, calyx teeth broadly ovate, vittae obscure, whereas *S. pauciflora* had fruits that were long ellipsoid, densely covered with sharp prickles, calyx teeth long-lanceolate, vittae 2 in the commissural side. *S. orthacantha* had narrowly ovoid fruit, densely covered with short, straight and sharp spines, and sometimes the base formed a thin layer, fruit ribs and furrows spinulose, narrowly lanceolate calyx teeth, while *S. brevispina* had the oblong ovoid to ovoid fruit with proximal end with degenerated-to-disappeared prickles, nearly smooth and distal end with prickles and formed a thin layer, usually with erose-spinulose ribs and furrows smooth or barely spinulose, linear to lanceolate calyx teeth, tip sharp. It can be seen that fruit micromorphology also supported that these three varieties should be considered as three independent species. The vittae was also one of the most important distinguishing features, such as *S. rubriflora*, *S. chinensis*, *S. caerulescens*, *S. pengshuiensis*, *S. pauciflora*, *S. lamelligera*, *S. oviformis*, *S. flavovirens* and *S. elata*, which possessed vittae, while other species had no vittae. The cell shapes of fruit furrows in *S. rubriflora*, *S. giraldii*, *S. flavovirens*, *S. tienmuensis* and *S. nanchuanensis* were polygonal, whereas *S. elongata*, *S. brevispina* and *S. oviformis* were circular. All these distinctive features can easily distinguish *Sanicula* members. Thus, it was noticed that prickles, fruit size, shape and vittae were the most distinctive features that can distinguish morphologically similar species of the genus *Sanicula*.

### 4.3. The Micromorphological Characteristics and Its Taxonomic Value of Pollen in the Genus Sanicula *L.*

The pollen is an important reproductive organ of plants and the pollen morphology, including pollen size, outer wall ornamentation, and germinal pore characteristics, which can reflect plant affinities to a certain extent, which has long been widely used in plant classification and phylogenetic studies [60,61]. In taxonomic studies of the family Apiaceae, the rich and diverse pollen micromorphological characteristics can also provide a valuable reference for exploring the phylogenetic relationships and evolutionary status between species [62,63,64,65,66,67]. For example, Xu et al. observed the pollen of seven *Physospermopsis* species and illustrated that *P. obtusiuscula* and *P. kingdon-wardii* can be distinguished based on their pollen shape because *P. obtusiuscula* was a relatively primitive rhombic-shaped pollen, while *P. kingdon-wardii* was a relatively evolved elliptical pollen [35]. Zhang et al. used light microscopy and scanning electron microscopy to observe the pollen morphology of *Ostericum* Hoffm. (Apiaceae) of eight species from seventeen populations. Their study showed that pollen morphology of *Ostericum* exhibited certain intraspecific stability in pollen shapes and exine ornamentations, while interspecies diversities in aperture types, central protrudents and length, density and appendages of exine ornamentation. In addition, they also found that the volume index, germinal furrows, and the outer wall decoration and wax stripes of *O. atropurpureum* and *O. citriodorum* were very similar, which can clearly distinguish them from other *Ostericum* members [65].

In this study, we observed that the location of germination pores and exine ornamentation of these twenty taxa were quite similar, indicating that the genus *Sanicula* was a natural unit, which was consistent with the pollen studies of Yang et al. [22]. However, the pollen shape and size of twenty *Sanicula* taxa were different. For example, *S. caerulescens*, *S. pauciflora* and *S. nanchuanensis* had equatorially constricted pollen, and thus they can be easily identified from others by the pollen shape. *S. flavovirens* had the largest pollen size (57.18 × 23.59 μm), while *S. caerulescens* had the smallest pollen size (32.83 × 14.09 μm), based on which they can be identified from each other by the pollen size. *S. pengshuiensis* had the largest P/E ratio of 2.35, whereas *S. oviformis* had the smallest P/E ratio of 1.58, which can identify them from other *Sanicula* members by the pollen P/E ratio. We also found that three varieties had distant affinities with their type varieties from pollen. For example, *S. giraldii* had a subrectangular equatorial view and a suborbicular polar view, while an ellipsoid equatorial view and trilobate circular polar view were observed in *S. subgiraldii. S. tienmuensis* had an ellipsoid equatorial view, whereas *S. pauciflora* had an equatorially constricted equatorial view. *S. orthacantha* had a suborbicular polar view, whereas *S. brevispina* had a triangular–circular polar view. Thus, pollen micromorphological also robustly supported that these three varieties should be treated as three independent species. Overall, the pollen micromorphological characteristics (such as pollen shape, size, P/E ratio, equatorial view and polar view) can also be used as an additional evidence to discriminate the complicated species of this genus.

## 5. Conclusions

In this study, we newly obtained leaf materials from twenty-two *Sanicula* members, fruit and pollen materials from twenty *Sanicula* members, covering about 90% of the Chinese samples, and performed comprehensively micromorphological analyses of this taxonomically difficult genus. Our study revealed the micromorphology of leaf epidermis, fruit and pollen of the genus in detail. In addition, we also evaluated the potential of micromorphology to distinguish species with taxonomic difficulties within the genus. Furthermore, we also provided more additional evidence for exploring the interspecific relationship of the genus. In summary, our study will be beneficial to future taxonomic studies of the genus *Sanicula*.

## Figures and Tables

**Figure 1 plants-13-01635-f001:**
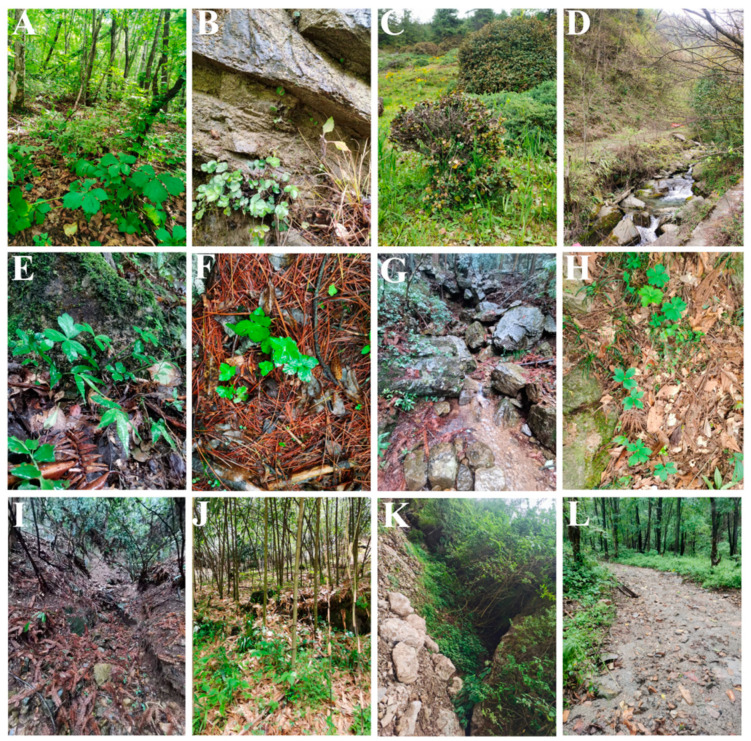
The habit of the *Sanicula* plants. (**A**): mixed forests on mountain slopes; (**B**): rocky walls; (**C**): alpine scrub; (**D**): grassy places on stream banks; (**E**): rock crevices; (**F**): inside the humus; (**G**): beside the stream; (**H**): crack of a stone slab; (**I**): wet valleys; (**J**): bamboo forests; (**K**): shady slope; (**L**): shady wet places.

**Figure 2 plants-13-01635-f002:**
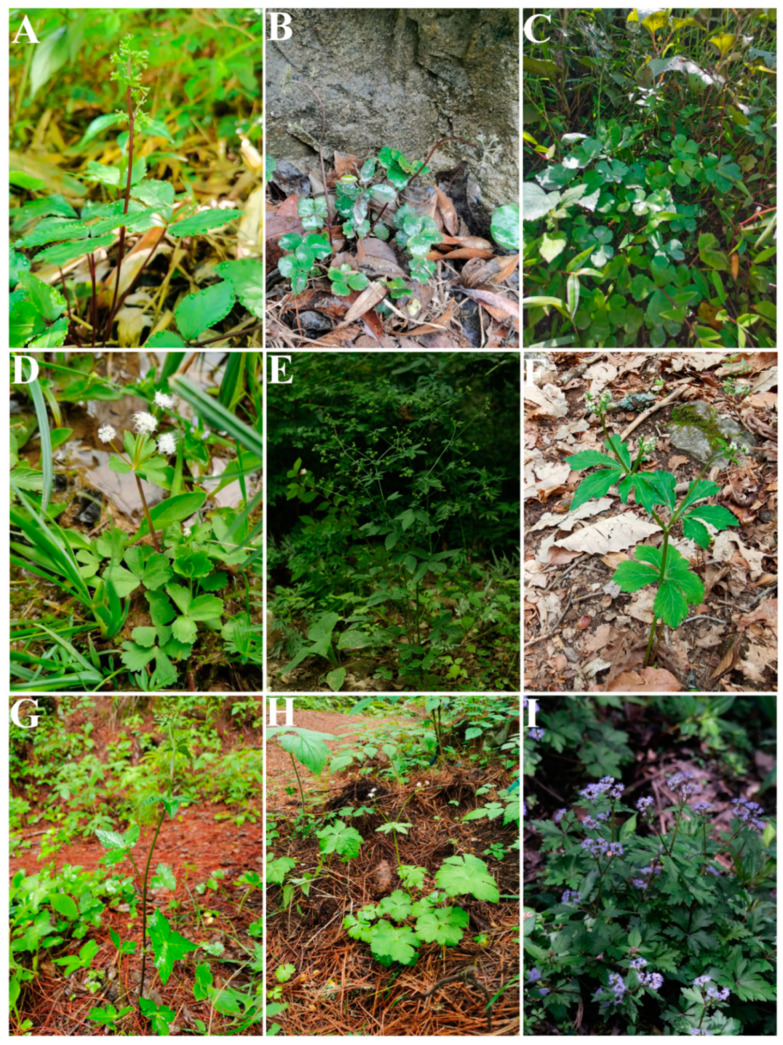
The morphology of the *Sanicula* plants. (**A**): *S. caerulescens*; (**B**): *S. oviformis*; (**C**): *S. rugulosa*; (**D**): *S. hacquetioides*; (**E**): *S. chinensis*; (**F**): *S. giraldii*; (**G**): *S. astrantiifolia*; (**H**): *S. serrata*; (**I**): *S. orthacantha*.

**Figure 3 plants-13-01635-f003:**
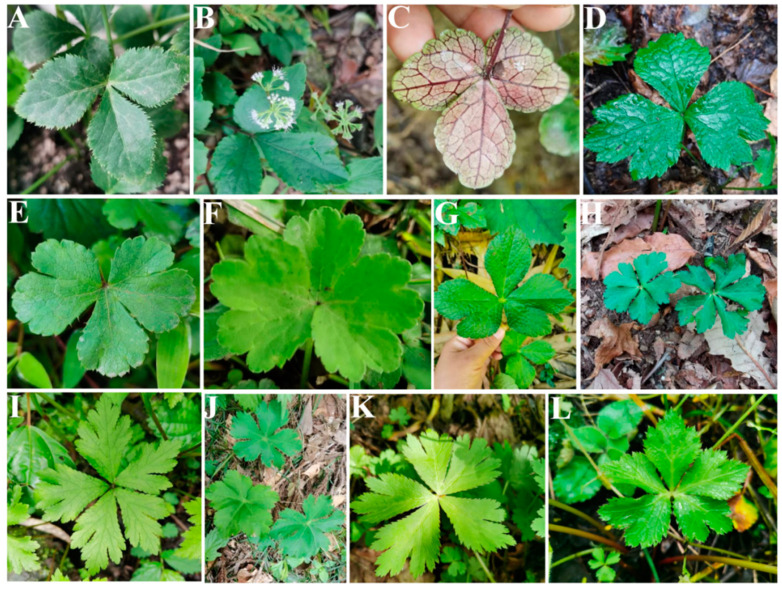
The basal leaves of *Sanicula* plants. (**A**–**C**): trifoliolate; (**D**–**G**): palmately 3-parted; (**H**–**L**): palmately 3-5-parted.

**Figure 4 plants-13-01635-f004:**
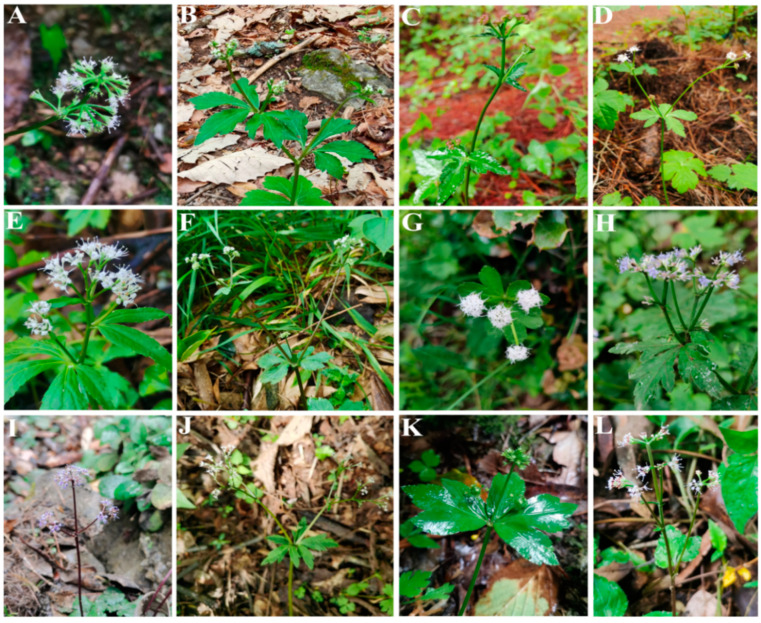
The inflorescence of *Sanicula* plants. (**A**): *S. caerulescens*; (**B**): *S. giraldii*; (**C**): *S. astrantiifolia*; (**D**): *S. serrata*; (**E**): *S. brevispina*; (**F**): *S*. *subgiraldii*; (**G**): *S. hacquetioides*; (**H**): *S. orthacantha*; (**I**): *S. oviformis*; (**J**): *S. elongata*; (**K**): *S. flavovirens*; (**L**): *S. pengshuiensis*.

**Figure 5 plants-13-01635-f005:**
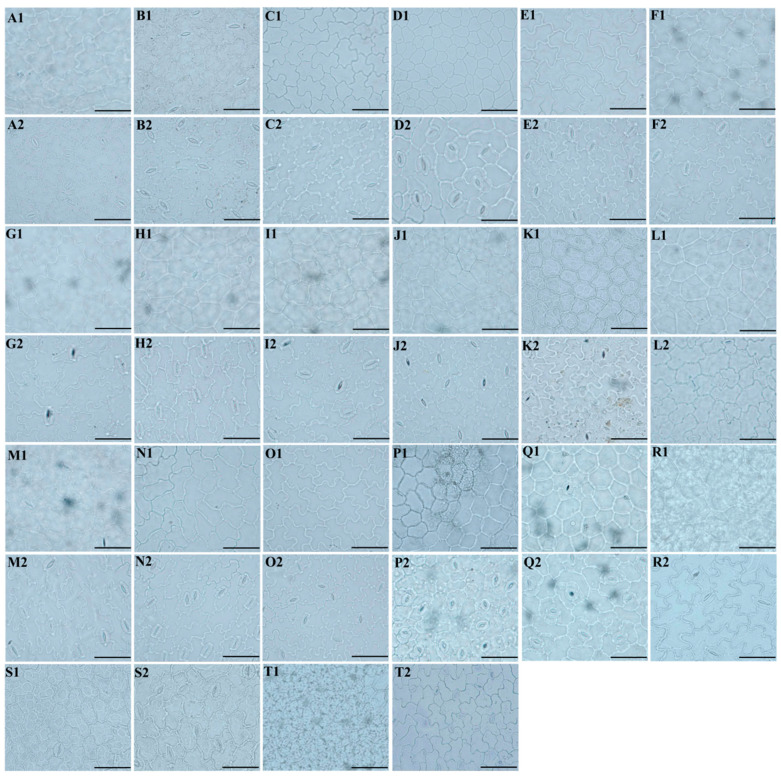
Leaf epidermal characteristics of *Sanicula* taxa observed with a light microscope. (**A**): *S. rubriflora*; (**B**): *S. rugulosa*; (**C**): *S. astrantiifolia*; (**D**): *S. chinensis*; (**E**): *S. giraldii*; (**F**): *S. caerulescens*; (**G**): *S. serrata*; (**H**): *S. elongata*; (**I**): *S. brevispina*; (**J**): *S*. *subgiraldii*; (**K**): *S. pengshuiensis*; (**L**): *S. pauciflora*; (**M**): *S. hacquetioides*; (**N**): *S. orthacantha*; (**O**): *S. lamelligera*; (**P**): *S. oviformis*; (**Q**): *S. tienmuensis*; (**R**): *S. flavovirens*; (**S**): *S. elata*; (**T**): *S. nanchuanensis*. 1: upper epidermis; 2: lower epidermis. Scale bar = 50 μm.

**Figure 6 plants-13-01635-f006:**
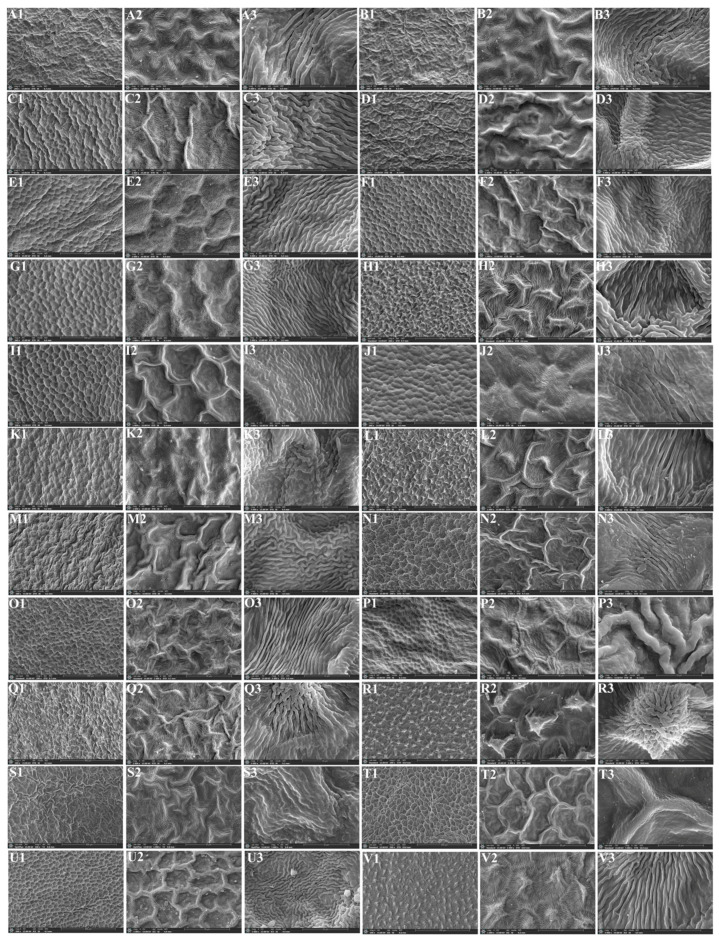
The micromorphological characteristics of the upper leaf epidermis observed with scanning electron microscope. (**A**): *S. rubriflora*; (**B**): *S. rugulosa*; (**C**): *S. astrantiifolia*; (**D**): *S. chinensis*; (**E**): *S. giraldii*; (**F**): *S. caerulescens*; (**G**): *S. serrata*; (**H**): *S. elongata*; (**I**): *S. brevispina*; (**J**): *S. subgiraldii*; (**K**): *S. pengshuiensis*; (**L**): *S. pauciflora*; (**M**): *S. hacquetioides*; (**N**): *S. orthacantha*; (**O**): *S. lamelligera*; (**P**): *S. oviformis*; (**Q**): *S. tienmuensis*; (**R**): *S. flavovirens*; (**S**): *S. elata*; (**T**): *S. nanchuanensis*; (**U**): *S. petagnioides*; (**V**): *S. tuberculata*. 1: 200×; 2: 1000×; 3: 5000×.

**Figure 7 plants-13-01635-f007:**
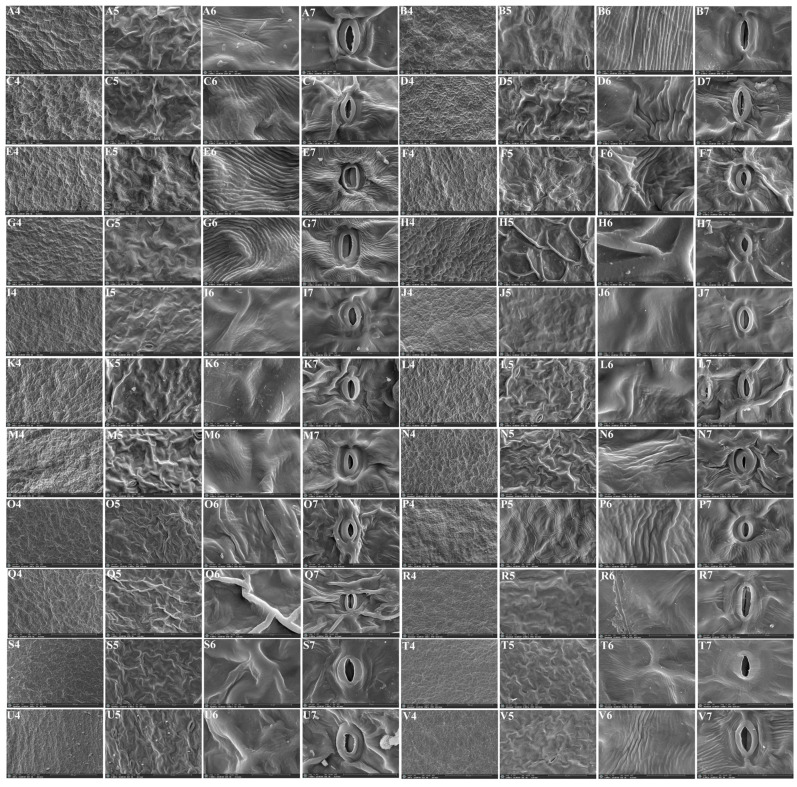
The micromorphological characteristics of the lower leaf epidermis observed with scanning electron microscope. (**A**): *S. rubriflora*; (**B**): *S. rugulosa*; (**C**): *S. astrantiifolia*; (**D**): *S. chinensis*; (**E**): *S. giraldii*; (**F**): *S. caerulescens*; (**G**): *S. serrata*; (**H**): *S. elongata*; (**I**): *S. brevispina*; (**J**): *S. subgiraldii*; (**K**): *S. pengshuiensis*; (**L**): *S. pauciflora*; (**M**): *S. hacquetioides*; (**N**): *S. orthacantha*; (**O**): *S. lamelligera*; (**P**): *S. oviformis*; (**Q**): *S. tienmuensis*; (**R**): *S. flavovirens*; (**S**): *S. elata*; (**T**): *S. nanchuanensis*; (**U**): *S. petagnioides*; (**V**): *S. tuberculata*. 4: 200×; 5: 1000×; 6: 5000×; 7: stoma.

**Figure 8 plants-13-01635-f008:**
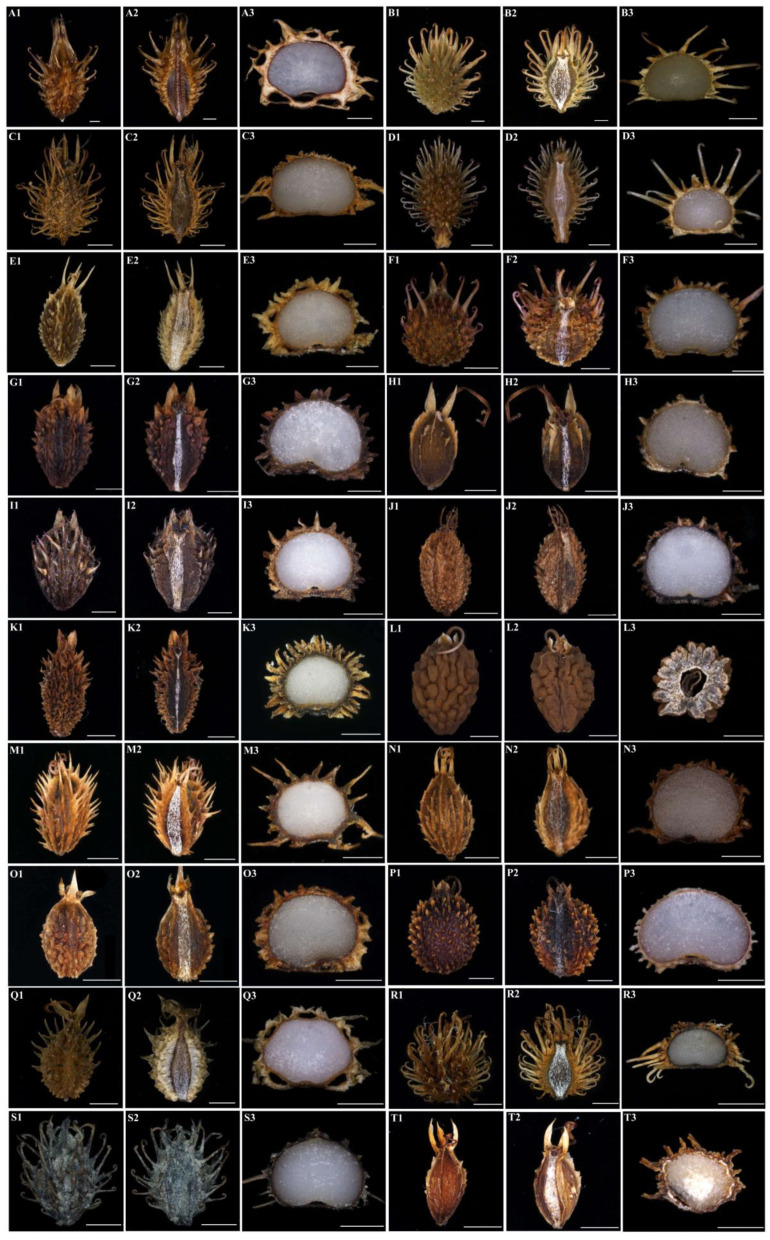
The morphological characteristics of the *Sanicula* fruit. (**A**): *S. rubriflora*; (**B**): *S. astrantiifolia*; (**C**): *S. chinensis*; (**D**): *S. giraldii*; (**E**): *S. caerulescens*; (**F**): *S. serrata*; (**G**): *S. elongata*; (**H**): *S. brevispina*; (**I**): *S*. *subgiraldii*; (**J**): *S. pengshuiensis*; (**K**): *S. pauciflora*; (**L**): *S. hacquetioides*; (**M**): *S. orthacantha*; (**N**): *S. lamelligera*; (**O**): *S. oviformis*; (**P**): *S. tienmuensis*; (**Q**): *S. flavovirens*; (**R**): *S. elata*; (**S**): *S. rugulosa*; (**T**): *S. nanchuanensis*. 1: dorsal side views of fruits; 2: commissural side views of fruits; 3: transverse section. Scale bar = 0.5 mm.

**Figure 9 plants-13-01635-f009:**
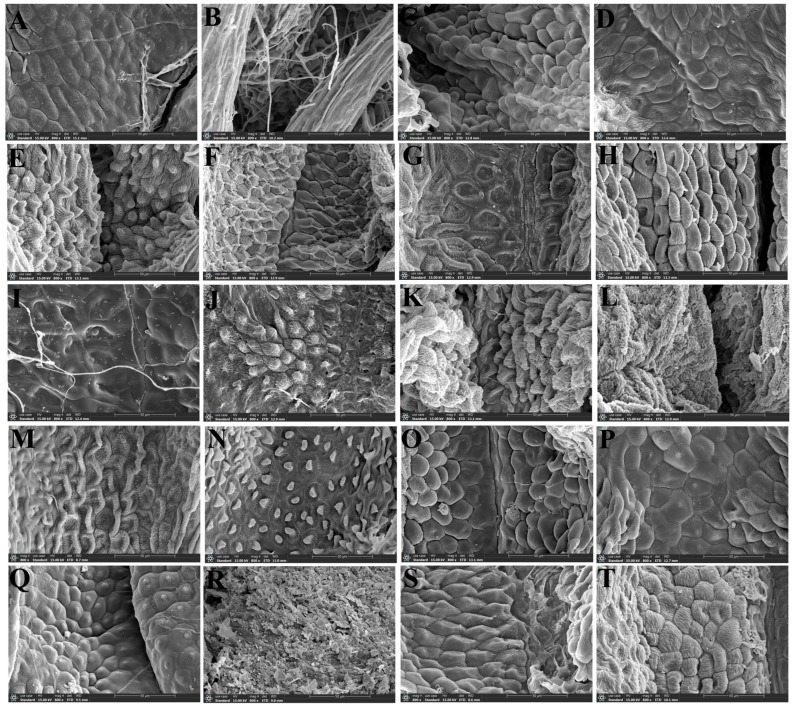
The micromorphological features of the *Sanicula* fruit observed with scanning electron microscope. (**A**): *S. rubriflora*; (**B**): *S. astrantiifolia*; (**C**): *S. chinensis*; (**D**): *S. giraldii*; (**E**): *S. caerulescens*; (**F**): *S. serrata*; (**G**): *S. elongata*; (**H**): *S. brevispina*; (**I**): *S*. *subgiraldii*; (**J**): *S. pengshuiensis*; (**K**): *S. pauciflora*; (**L**): *S. hacquetioides*; (**M**): *S. orthacantha*; (**N**): *S. lamelligera*; (**O**): *S. oviformis*; (**P**): *S. tienmuensis*; (**Q**): *S. flavovirens*; (**R**): *S. rugulosa*; (**S**): *S. elata*; (**T**): *S. nanchuanensis*.

**Figure 10 plants-13-01635-f010:**
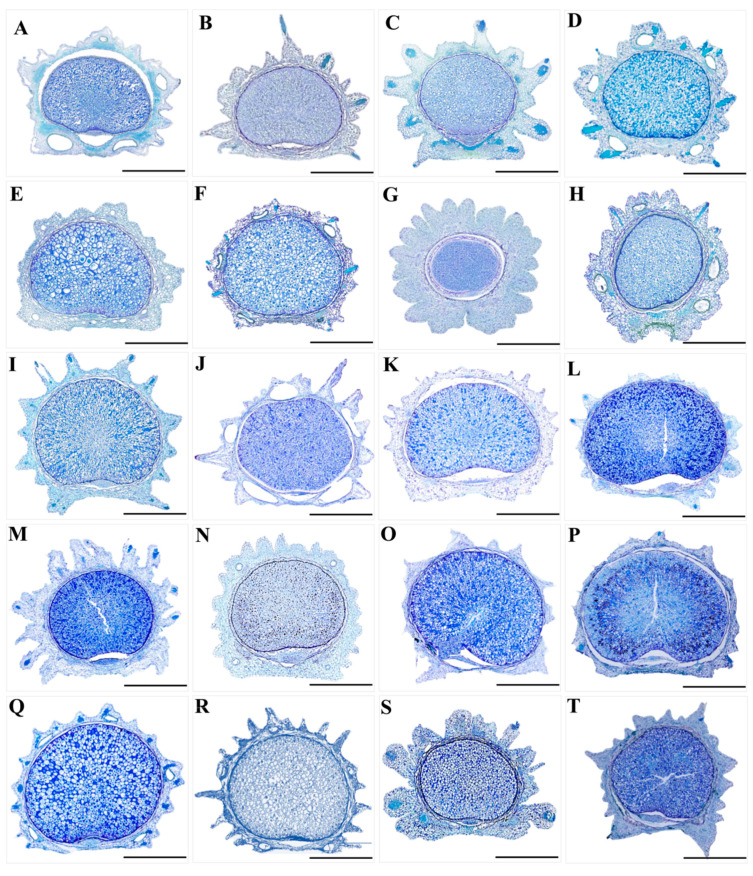
The micromorphological features of paraffin slices of *Sanicula* fruits. (**A**): *S. rubriflora*; (**B**): *S. astrantiifolia*; (**C**): *S. giraldii*; (**D**): *S. caerulescens*; (**E**): *S. elongata*; (**F**): *S. pengshuiensis*; (**G**): *S. hacquetioides*; (**H**): *S. oviformis*; (**I**): *S. orthacantha*; (**J**): *S. flavovirens*; (**K**): *S. tienmuensis*; (**L**): *S. serrata*; (**M**): *S. subgiraldii*; (**N**): *S. pauciflora*; (**O**): *S. brevispina*; (**P**): *S. rugulosa*; (**Q**): *S. lamelligera*; (**R**): *S. chinensis*; (**S**): S. *elata*; (**T**): *S. nanchuanensis*. Scale bar = 0.5 mm.

**Figure 11 plants-13-01635-f011:**
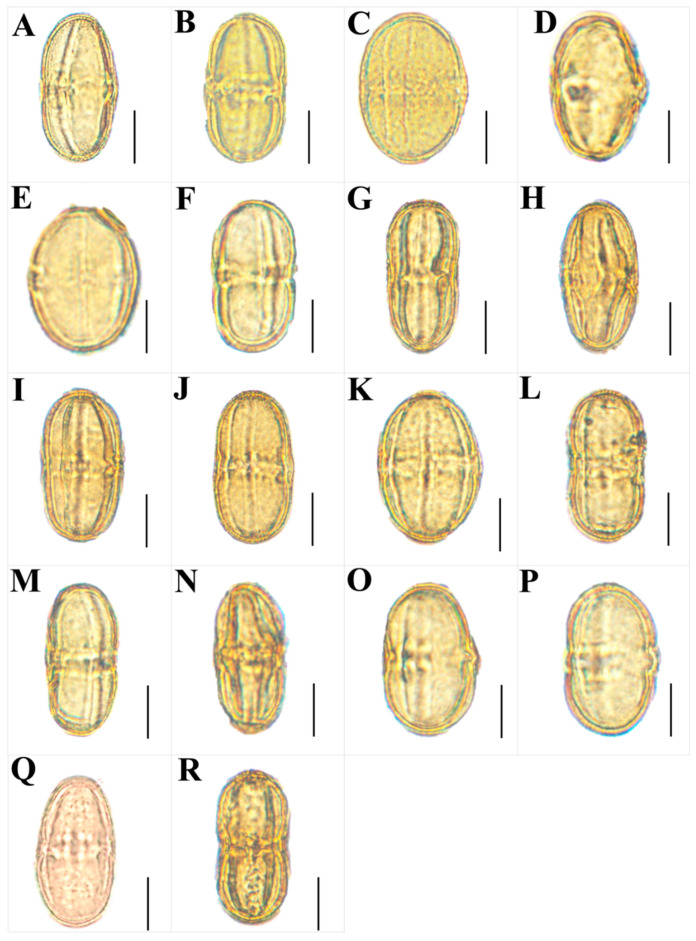
The micromorphological characteristics of pollen for *Sanicula* plants observed with a light microscope. (**A**): *S. rubriflora*; (**B**): *S. rugulosa*; (**C**): *S. astrantiifolia*; (**D**): *S. chinensis*; (**E**): *S. giraldii*; (**F**): *S. caerulescens*; (**G**): *S. serrata*; (**H**): *S. elongata*; (**I**): *S. brevispina*; (**J**): *S. subgiraldii*; (**K**): *S. pengshuiensis*; (**L**): *S. pauciflora*; (**M**): *S. hacquetioides*; (**N**): *S. orthacantha*; (**O**): *S. lamelligera*; (**P**): *S. tienmuensis*; (**Q**): *S. elata*; (**R**): *S. nanchuanensis*. Scale bar = 10 μm.

**Figure 12 plants-13-01635-f012:**
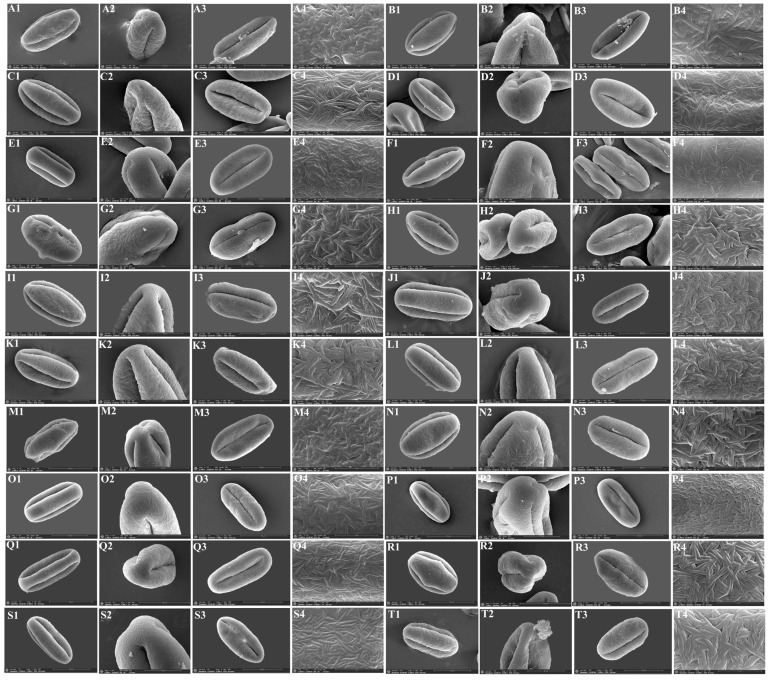
The micromorphological characteristics of pollen for *Sanicula* plants observed with a scanning electron microscope. (**A**): *S. rubriflora*; (**B**): *S. rugulosa*; (**C**): *S. astrantiifolia*; (**D**): *S. chinensis*; (**E**): *S. giraldii*; (**F**): *S. caerulescens*; (**G**): *S. serrata*; (**H**): *S. elongata*; (**I**): *S. brevispina*; (**J**): *S. subgiraldii*; (**K**): *S. pengshuiensis*; (**L**): *S. pauciflora*; (**M**): *S. hacquetioides*; (**N**): *S. orthacantha*; (**O**): *S. lamelligera*; (**P**): *S. tienmuensis*; (**Q**): *S. elata*; (**R**): *S. oviformis*; (**S**): *S. flavovirens*; (**T**): *S. nanchuanensis.* 1: equatorial view; 2: polar view; 3: germinal furrow; 4: exine ornamentation of equatorial view.

**Table 1 plants-13-01635-t001:** The morphological characteristics of fruits for *Sanicula* plants.

Taxon	Shape	Size (mm)	Fruit Surface	Calyx Teeth	Endosperm on Commissural Side	Vittae
*S. rubriflora*	Ovoid or ovoid–globose	6.97 × 2.68	proximal part covered with scales, distal part covered with yellow uncinate bristles	ovate–lanceolate	slightly concave	5
*S. astrantiifolia*	Obovate or subglobose	1.89 × 0.90	proximal end with short bristles, distal end with uncinate bristles, bristles yellow or purple-red	linear–lanceolate	flat	Obscure
*S. chinensis*	Ovoid–globose	3.28 × 1.37	bristles uncinate above, dilated at base	linear	flat	5
*S. giraldii*	Narrowly ovoid	3.86 × 1.74	densely covered with developed yellow or purplish red uncinate bristles, long and hard	ovate and small, tip mucronate	flat	Obscure
*S. caerulescens*	Globose or ellipsoid	2.35 × 0.68	short and straight spinous-bristles usually fused at the base forming a thin tier	linear–lanceolate	slightly concave	5
*S. serrata*	Ovoid or ovoid–globose	2.08 × 1.60	proximal part covered with scales, distal part covered with slightly uncinate bristles, bristles pale yellow or purplish red	ovate	slightly concave	Obscure
*S. elongata*	Ovoid	2.39 × 1.11	densely covered with pale yellow scales	narrow–ovate	slightly concave	Obscure
*S. brevispina*	Oblong ovoid to ovoid	1.95 × 1.25	proximal end with degenerated to disappeared the prickles, nearly smooth, distal end with prickles and formed a thin layer; usually with erose-spinulose ribs and furrows smooth or barely spinulose	linear to lanceolate	slightly concave	Obscure
*S. subgiraldii*	Broadly ovate	4.01 × 2.51	rarely covered with purplish-red short bristles, proximal end with tubercles, obscure, distal end with uncinate bristles or straight	broadly ovate and large	slightly concave	Obscure
*S. pengshuiensis*	Ellipsoid	2.38 × 0.77	bristles in regular rows in furrows, ribs glabrous, stout and prominent	linear	flat	5
*S. pauciflora*	Long ellipsoid	2.72 × 0.83	densely covered with sharp prickles	long–lanceolate	flat	vittae 2 in commissural side
*S. hacquetioides*	Ovoid–globose	2.81 × 2.08	covered with scales and tubercles, but never spinulose	broadly ovate or obovate	flat	Obscure
*S. orthacantha*	Narrowly ovoid	2.02 × 0.86	short and straight spines	narrowly lanceolate	flat	Obscure
*S. lamelligera*	Long-ovoid	1.97 × 0.72	short and straight spines, never uncinate, fused at the base forming a thin tier	linear	slightly concave	5
*S. oviformis*	Ovoid	2.13 × 1.44	densely short and straight-spinulose; ribs prominent	linear–lanceolate	slightly concave	5
*S. tienmuensis*	Subglobose	2.52 × 1.82	covered with short and obtuse prickles, slight formed scales and tubercles	broadly ovate	slightly concave	Obscure
*S. flavovirens*	Ovoid–globose	3.26 × 1.79	proximal part covered with tubercles, distal part covered with yellow straight prickles	triangular–lanceolate, tip mucronate	flat	5
*S. elata*	Ovoid–globose	2.27 × 1.35	densely covered with yellow uncinate bristles	narrowly lanceolate	flat	5
*S. rugulosa*	Ellipsoid	2.41 × 1.45	densely covered with uncinate bristles	narrowly lanceolate	slightly concave	Obscure
*S. nanchuanensis*	Ellipsoid	2.74 × 1.65	proximal part covered with scales, rarely formed bristles	triangular–lanceolate	flat	Obscure

## Data Availability

Data are contained within the article and Appendix A.

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
