# Peer review of "The Micromorphology and Its Taxonomic Value of the Genus Sanicula L. in China (Apiaceae)"

_plants, 2024, doi:10.3390/plants13121635_

Round 1
Reviewer 1 Report
Comments and Suggestions for Authors
It's a very valuable manuscript, based on huge material. All used methods are proper, and conclusions are correct. I strongly recommend publishing this paper.
I have a few comments:
1. All keywords are in the title, I suggest using others.
2. The photos are poor quality in Figure 1. Should be changed.
3. There aren't explanations for numbers 1, 2, 3... in Figure 6 and 4,5,6,7 in Figure 7, 1,2,3 in Figure 8, 1,2,3,4 in Figure 12.
4. Letters should be white in Figure 9. They are invisible.
5. There aren't explanations of scale bars under all Figures.
Author Response
Reviewers' comments:
Reviewer #1:
It’s a very valuable manuscript, based on huge material. All used methods are proper, and conclusions are correct. I strongly recommend publishing this paper.
I have a few comments:
- All keywords are in the title, I suggest using others.
Response: Thanks for your suggestions. We have used other keywords (Line 40).
- The photos are poor quality in Figure 1. Should be changed.
Response: Thanks, we are so sorry that the image is poor in Figure 1, we have improved the resolution of Figure 1 and re-uploaded it in the revised version.
- There aren’t explanations for numbers 1, 2, 3... in Figure 6 and 4,5,6,7 in Figure 7, 1,2,3 in Figure 8, 1,2,3,4 in Figure 12.
Response: Thanks for your comments. We have added the explanations for numbers 1, 2, 3... in Figure 6 and 4,5,6,7 in Figure 7, 1,2,3 in Figure 8, 1,2,3,4 in Figure 12 (Line 308, 325, 356, 424).
- Letters should be white in Figure 9. They are invisible.
Response: Thanks. We have re-written the letters in Figure 9 and re-uploaded it in the revised version.
- There aren't explanations of scale bars under all Figures.
Response: Thanks for your careful comments and suggestions. We have added the scale bars under all Figures.

Reviewer 2 Report
Comments and Suggestions for Authors
The paper is interesting and is a very detailed study of 90% of the micromorphology of species of the Sanicula genus in China (how many are missing? perhaps they are very rare or difficult to find). However, based on the results, it would be interesting to put some order among the Sanicula species and describe the affinities between the different species on the basis of the observed characters. Perhaps you have another article in preparation that tries to make some phylogenetic hypotheses on the basis of the characters detected and the literature. It would be a shame to collect all this data and leave the synthesis that can be drawn from it to other researchers. In the attached text I highlighted some typos, above all check the italic writing of the names of the species which in many captions does not appear correct.

Author Response
Reviewer #2:
The paper is interesting and is a very detailed study of 90% of the micromorphology of species of the Sanicula genus in China (how many are missing? perhaps they are very rare or difficult to find). However, based on the results, it would be interesting to put some order among the Sanicula species and describe the affinities between the different species on the basis of the observed characters. Perhaps you have another article in preparation that tries to make some phylogenetic hypotheses on the basis of the characters detected and the literature. It would be a shame to collect all this data and leave the synthesis that can be drawn from it to other researchers. In the attached text I highlighted some typos, above all check the italic writing of the names of the species which in many captions does not appear correct.
Response: Thanks for your comments. Two species are missing because they are difficult to find. We have described the affinities between the different species on the basis of the observed characters in the discussion section (Lines 485-494, 542-558, 598-606). We also checked and revised the the names of the species which in many captions does not appear correct.
1.perhaps it is bettere to say "habit".
Response: Thanks. We have showed the habit of the Sanicula plants In the Figure 1.
- put plant species names in italics here and check the following captions.
Response: Thanks. We have checked and put the plant species names in italics in the text.

Reviewer 3 Report
Comments and Suggestions for Authors
Dear authors,
This manuscript constitutes an interesting study concerning the micromorphology and its taxonomic value of the genus Sanicula L. (Apiaceae). The overall idea is interesting, and the methodology is well described. The manuscript, in general, is clear and well-structured. In the methodology is not written the reason for which these 22 taxa examined have been selected. It is mentioned that there are 19 species and five varieties of the genus distributed in China. In my opinion the study should concern only species level of the genus and not varieties. So, the study could concern the 19 species as representing the diversity of the genus in China, and the title should be changed accordingly as also different parts of the manuscript, Figures and Supplementary Tables.
There are some more points, as they are described below, that could be improved.
I hope that my comments will be helpful.
Title of the manuscript: In my opinion the title could be more informative and possibly it could be changed to “The micromorphology and its taxonomic value of the genus Sanicula L. in China (Apiaceae).”.
Page 2, 2nd paragraph, line 7: Please write correctly this reference “For example, Ma et al. observed”.
Figures 2, 4, 5, 9 and 10: Please write scientific names of the plants (genus and species) in italics
Page 17, 4.2, 1st paragraph, line 5: Please change “Liu et al. (2006)” to a Reference number as also “Kljuykov et al. (2021)” on line 9 of the same paragraph
Supplementary files, Table S2: Please check. In Table S2, there are 3 Tables, 2 of which under the same renumeration (3.5, 3.5), they should also be in different sheets or pages:
Table 3.5 The features of upper leaf epidermis for Sanicula plants observed with LM.,
Table 3.5 The features of lower leaf epidermis for Sanicula plants observed with LM.
Table 3.6 The features of leaf epidermis for Sanicula plants observed with SEM.
I hope that my comments will be helpful.
Author Response
Reviewer #3:
Dear authors,
This manuscript constitutes an interesting study concerning the micromorphology and its taxonomic value of the genus Sanicula L. (Apiaceae). The overall idea is interesting, and the methodology is well described. The manuscript, in general, is clear and well-structured. In the methodology is not written the reason for which these 22 taxa examined have been selected. It is mentioned that there are 19 species and five varieties of the genus distributed in China. In my opinion the study should concern only species level of the genus and not varieties. So, the study could concern the 19 species as representing the diversity of the genus in China, and the title should be changed accordingly as also different parts of the manuscript, Figures and Supplementary Tables. There are some more points, as they are described below, that could be improved. I hope that my comments will be helpful.
Response: Thanks for your comments. The reason why we selected 22 taxa was these species have similar morphology and species identification was difficult. We tried to find out their identifying features from micromorphology. We also added the reason in the revised version (Lines 94-95). The three varieties involved in this study should be regarded as three independent species, and our molecular phylogenetic article confirmed that these three varieties should be treated as three independent species. The three varieties included in this study were aimed at finding micromorphological evidence for their status as three independent species and we have also added these contents in the discussion section (Lines 485-494, 542-558, 598-606).
- Title of the manuscript: In my opinion the title could be more informative and possibly it could be changed to “The micromorphology and its taxonomic value of the genus Sanicula L. in China (Apiaceae).”.
Response: Thanks for your comments and suggestions, we have revised the manuscript title according to your comments in the revised version.
- Page 2, 2nd paragraph, line 7: Please write correctly this reference “For example, Ma et al. observed”.
Response: Thanks, we have written correctly this reference in the revised text (Line 76).
- Figures 2, 4, 5, 9 and 10: Please write scientific names of the plants (genus and species) in italics.
Response: Thank you. We have revised the scientific names of the plants (genus and species) in italics of Figures 2, 4, 5, 9 and 10.
- Page 17, 4.2, 1st paragraph, line 5: Please change “Liu et al. (2006)” to a Reference number as also “Kljuykov et al. (2021)” on line 9 of the same paragraph.
Response: Thanks for your comments. We have changed “Liu et al. (2006)” to a Reference number as also “Kljuykov et al. (2021)” on line 9 of the same paragraph (Line 504, 508).
- Supplementary files, Table S2: Please check. In Table S2, there are 3 Tables, 2 of which under the same renumeration (3.5, 3.5), they should also be in different sheets or pages:
Table 3.5 The features of upper leaf epidermis for Sanicula plants observed with LM.,
Table 3.5 The features of lower leaf epidermis for Sanicula plants observed with LM.
Table 3.6 The features of leaf epidermis for Sanicula plants observed with SEM.
I hope that my comments will be helpful.
Response: Thanks. We have carefully examined all Supplementary files in the whole text and revised the incorrect files.
The features of upper leaf epidermis for Sanicula plants observed with LM in Table S2-1.
The features of lower leaf epidermis for Sanicula plants observed with LM in Table S2-2.
The features of upper leaf epidermis for Sanicula plants observed with SEM in Table S3-1.
The features of lower leaf epidermis for Sanicula plants observed with SEM in Table S3-2.
The pollen morphology of Sanicula plants observed with SEM in Table S4.

Round 2
Reviewer 2 Report
Comments and Suggestions for Authors
The article has been deeply revised by the authors who have explored and better explained the methodology used; they have also better detailed the morphological characteristics from which important taxonomic information can be drawn. the article can therefore be published on Plants. I repeat that an identification key would have been useful. In the attached file I highlight a few typos

Comments on the Quality of English LanguageAuthor Response
The article has been deeply revised by the authors who have explored and better explained the methodology used; they have also better detailed the morphological characteristics from which important taxonomic information can be drawn. the article can therefore be published on Plants. I repeat that an identification key would have been useful. In the attached file I highlight a few typos.
Response: Thanks for your positive comments. We have carefully checked the whole text and corrected a number of minor flaw in our revised version according to your attached file. The revised portions are tracked with blue words in the text.

Reviewer 3 Report
Comments and Suggestions for Authors
Dear authors,
The revised version of your manuscript is much improved. I do not have other comments or suggestions.
Author Response
Dear authors,
The revised version of your manuscript is much improved. I do not have other comments or suggestions.
Response: Thanks for your positive comments.